# Knowledge, attitude and practice (KAP) and risk factors on dengue fever among children in Brazil, Fortaleza: A cross-sectional study

**Yang Zhang**[1], **Monica Zahreddine**[2], **Kellyanne Abreu**[3], **Mayana Azevedo Dantas**[3], **Katia Charland**[2], **Laura Pierce**[2], **Valéry Ridde**[4], **Kate Zinszer**[2,5]*

**1** School of Population and Global Health, McGill University, Montreal, Quebec, Canada, **2** Center for Public Health Research, University of Montreal, Montreal, Quebec, Canada, **3** State University of Ceará, Ceará, Brazil, **4** Institute for Research on Sustainable Development, CEPED (IRD-Université de Paris), Université de Paris, ERL INSERM SAGESUD, Paris, France, **5** School of Public Health, University of Montreal, Montreal, Quebec, Canada

* kate.zinszer@umontreal.ca

**Data Availability Statement:** Requests for de-identified or anonymized data should be sent to the research ethics board at Université de Montréal,

## Abstract

### Background

Dengue fever is a mosquito-borne viral disease that is associated with four serotypes of the dengue virus. Children are vulnerable to infection with the dengue virus, particularly those who have been previously infected with a different dengue serotype. Sufficient knowledge, positive attitudes, and proper practices (KAP) are essential for dengue prevention and control. This study aims to estimate the dengue seropositivity for study participants and to examine the association between households' dengue-related knowledge, attitudes, and practices (KAP), and children's risk of dengue seropositivity, while accounting for socioeconomic and demographic differences in Brazil.

### Methodology/Principal findings

This analysis was based on a cross-sectional study from Fortaleza, Brazil between November 2019, and February 2020. There were 392 households and 483 participant children who provided a sample of sufficient quality for serological analysis. The main exposure was a household's dengue-related knowledge, attitudes, and practices, assessed through a questionnaire to construct a composite KAP score categorized into three levels: low, moderate, and high. The main outcome is dengue immunoglobulin G(IgG) antibodies, collected using dried blood spots and assessed with Panbio Dengue IgG indirect ELISA (enzyme-linked immunosorbent assays) test commercial kits.

The estimated crude dengue seroprevalence among participating children (n = 483) was 25%. Five percent of households (n = 20) achieved a score over 75% for KAP, sixty-nine percent of households (n = 271) scored between 50% and 75%, and twenty-six percent of households (n = 101) scored lower than 50%. Each KAP domain was significantly and positively associated with the others. The mean percentage scores for the three domains are 74%, 63%, and 39% respectively. We found high household KAP scores were associated

CERSES (email: cerses@umontreal.ca) to ensure that data is shared in accordance with participant consent. Participants were only asked for limited consent to use of the data in related studies conducted by the project investigators and their students, subject to the approval by a Research Ethics Board.

**Funding:** This research was funded by the Canadian Institutes of Health Research, through the Project Grant program (201803PJT-400444-RC2-CFCA-120159). The funders had no role in study design, data collection and analysis, decision to publish, or preparation of the manuscript.

**Competing interests:** The authors have declared that no competing interests exist.

with an increased adjusted relative risk (aRR) of seropositivity (aRR: 2.11, 95% CI: 1.11–4.01, p = 0.023). Household adult respondents' education level of elementary school or higher was negatively associated with children's risk of being seropositive (aRR: 0.65, 95% CI: 0.48–0.87, p = 0.005). The risk of seropositivity in older children (6–12 years old) was over 6 times that of younger children (2–5 years old) (aRR: 6.08, 95% CI: 3.47–10.64, p<0.001). Children living in households with sealed water tanks or no water storage had a lower risk of being seropositive (aRR: 0.73, 95% CI: 0.54–0.98, p = 0.035).

## Conclusions/Significance

Our results provide insight into the prevalence of dengue seropositivity in Fortaleza, Brazil in children, and certain demographic and socioeconomic characteristics associated with children's risk of being seropositive. They also suggest that KAP may not identify those more at-risk for dengue, although understanding and enhancing households' KAP is crucial for effective community dengue control and prevention initiatives.

## Author summary

Dengue fever has become an increasing threat to public health, with its global expansion and increased presence in dengue endemic countries. This study provides insight into the prevalence of dengue seropositivity in children in Fortaleza, Brazil, a city which has been a hotspot for arbovirus infections. We estimated seroprevalence for certain study population characteristics and identified characteristics that were associated with an increased risk of dengue seropositivity. We also explored the associations of a composite measure of knowledge, attitudes, and practice (KAP) with seropositivity and found an inverse relationship between KAP for dengue control and children's seropositivity. KAP could change due to previous infection experience, which is a potential limitation of using KAP as a potential predictor of dengue seropositivity in cross-sectional studies. Despite this, KAP remains useful for identifying gaps in knowledge, attitude, and practice that can be used to inform public health measures, such as education campaigns. Measures of social acceptability of interventions should also be considered for inclusion in similar studies, as it would provide an indication of likelihood of adoption of interventions, which provides additional insight for different dengue interventions.

## Introduction

Dengue fever is a mosquito-borne viral disease that is associated with four serotypes of the dengue virus [1]. Dengue is an expanding threat to public health globally and the leading cause of serious illness and death among children in several Asian and Latin American countries [1]. Children are vulnerable to infection with the dengue virus, particularly those who have been previously infected with a different dengue serotype [2]. Brazil, the largest country in South America, is experiencing increased dengue incidence, with over 1.5 million cases reported in 2019 [3]. A complex mixture of demographic and socio-economic factors influences the distribution and transmission of dengue [4], which include inadequate living conditions, immunologically naïve populations, global trade and population mobility, climate change, and the adaptive nature of the principal mosquito vectors *Aedes aegypti* and *Aedes albopictus* [5–8].

Given that dengue vaccines are not suitable nor available for all dengue-endemic countries and anti-viral medications are not available, control of the mosquito vectors and reducing human-vector contact are recognized as the primary strategies for dengue prevention [9]. *Aedes aegypti* and *Aedes albopictus* are opportunist mosquitoes adapted to urban environments [10]. Precipitation and open water storage create sites for breeding and aquatic stages of the mosquito's life cycle [11]. In several Brazilian cities, residents store clean water in open containers and water reservoirs creating ideal sites for *Aede*s within and near the households [12]. Poor quality housing and sanitation management with high population density are key determinants for the sustained propagation of dengue [13].

Measures of knowledge, attitudes, and practices (KAP) are common and used to identify knowledge gaps, evaluate awareness levels and assess behavioural patterns among members of the community, that may facilitate understanding and public health action [14]. High levels of KAP can empower individuals to participate in disease control and prevention programs [15]. Although there have been several studies assessing KAP and its relationship to dengue risk in different populations, little is known about the link between households' KAP, demographic and socio-economic characteristics and children's dengue seropositivity [16–18]. The objectives of this analysis were to estimate dengue seroprevalence in a pediatric study population and to examine the associations between households' dengue-related KAP and children's dengue seropositivity.

## Methods

### Ethics statement

Data collection for the study was approved by the University of Montreal's Comité d'éthique de la recherche en sciences et en santé (CERSES) and the Comité de Ética em Pesquisa da Universidade Estadual do Ceará (State University of Ceará, reference number 3.083.892). Approval for the study was obtained from the Ethical Review Committee for Scientific Research of Brazil's National Ethics Council for Research. All parents/guardians gave written informed consent (by signature or fingerprint) for their children to participate.

### Study design

This analysis is based on cross-sectional data collected as part of the baseline data collection of a larger international research project [19]. The original project was a pragmatic cluster randomized controlled trial (cRCT) in Fortaleza, Brazil, which aimed to evaluate a community mobilization intervention for dengue control. There are 3,020 census tracts (clusters) in Fortaleza and each cluster was enumerated and filtered by eligibility criteria. Eligible clusters were then categorized according to the dengue vulnerability index (DVI), based on the 2010 Census. To limit contamination among clusters, any selected cluster with common boundaries to previously selected clusters was excluded and replaced by the next cluster (from a ranked list based upon stratification criteria).

The cRCT was powered for the primary objective of comparing seroprevalence between those receiving the intervention and those not receiving the intervention [19]. The current study is part of a secondary objective, therefore a power analysis was not conducted. The cRCT planned to collect data from 68 clusters with a target recruitment of 5,848 children between 2 and 12 years old. For further detail, please see the original trial protocol in Zinszer et. al. 2020 [19]. Due to the Covid-19 pandemic and instability in the study region, the baseline data collection terminated before data from all clusters were gathered. The team was able to obtain the target sample size (n = 86 per cluster) in four clusters although there were eight clusters where this was not possible due to the field conditions during the time period of

November 2019 and February 2020. In the incomplete clusters, complete data was collected (e.g., questionnaire and DBS) on a smaller number of participants within these clusters.

## Eligibility criteria and study procedure

Households in selected clusters were eligible to participate if there were children between 2 to 12 years old living in the household, and if both the parent or legal guardian gave consent and the child gave assent for the data collection. There were two different informed consent forms (ICF) and one assent form. The first ICF was signed (or digit printed if not able to write) by the person responsible for the household and able to answer the household questionnaire (principal adult respondent). For each participating child, a specific ICF for serology blood collection was signed by the child's parent or guardian. Study objectives and the details of the procedures were explained in adapted language by trained study staff and an assent form was signed or fingerprinted by the participating child if they agreed to participate. Households and children with contaminated blood samples and/or with indeterminate serology results were excluded from this analysis.

Household surveys were administered to the principal adult respondent of each household in Portuguese by a trained interviewer during door-to-door visits. Household surveys included questions about living conditions, socioeconomic status, social capital, and basic health history of the participating children. Respondents were also asked questions related to dengue knowledge, attitudes, and practice (KAP).

## Knowledge, attitudes, and practice

The KAP module consisted of partially categorized questions as well as multiple choice, dichotomous and Likert scale responses (S1 Appendix). Five questions in the knowledge domain gauged awareness of dengue symptoms, modes of transmission, and techniques for dengue prevention. Five questions in the attitude domain assessed perceptions about dengue and opinions and acceptability of dengue prevention practices. There were six items in the practice domain related to the adoption of preventive practices and dengue treatment seeking behaviour. A scoring system was applied to create a KAP index. For the multiple choice and dichotomous questions, only correct responses were assigned positive scores (an incorrect response was given a score of zero). For Likert scale questions, the highest score was assigned for 'strongly agreed', 0 for 'strongly disagree' with gradations of 0.5 for 5-choice Likert scale questions. The maximum scores of 10, 8, and 10.5 points were applied to the knowledge, attitude, and practice domain, respectively. Households' total KAP score (maximum of 28.5 points) were categorized into groups: low (<15 points or <50% of maximum points), moderate (>15 or <22 points or between 50% to 75% of maximum score) or high (>22 points or >75% of the maximum score) [17,20,21].

## Seroprevalence test

Dried blood spot samples were collected from children by a finger prick performed by a trained nurse technician. All samples were placed on filter paper and were anonymized with a study ID code during collection. Three circles of 15 mm each were filled with capillary blood and allowed to dry at least 2 to 6 hours at room temperature, avoiding exposure to sunlight. Once dried, each card was packaged separately in a hermetic plastic bag containing silica desiccant. DBS samples were then refrigerated until assessment for quality control purposes, standards of sufficient quantity of blood or layering of blood drops is available elsewhere [22], and then stored in a freezer at -20°C. At the laboratory, three punches of 6 mm from the DBS were soaked in a 1.5 mL vial containing 250 μL of eluent solution, shaken overnight at 4°C to ensure

adequate serum elution. The next day, the eluates were centrifuged at 2,500 rpm for 2 minutes in a refrigerated microfuge and then transferred to a clean and sterile tube. Undiluted DBS eluates were treated as a 1:25 equivalent dilution of human serum, based on previous work that demonstrated that approximately three 6 mm circles contained 21 μL of dried blood, corresponding to approximately 10.5 μL of serum that were eluted in 250 μL of the eluent solution. Dengue seropositivity was defined as the detection of immunoglobulin G (IgG) antibodies to dengue antigen serotypes (1, 2, 3, and 4) using the Panbio Dengue IgG indirect ELISA (enzyme-linked immunosorbent assays), a commercial capture ELISA with high specificity and sensitivity, and excellent correlation with Hemagglutination Inhibition (HAI) assay [23,24]. To ensure high-quality data collection, a field supervisor was present for both blood collection and questionnaire administration.

## Statistical analyses

The primary outcome of this study was dengue seropositivity (yes/no). Seroprevalence with 95% confidence intervals were estimated using a single-proportion z-test. We conducted Chi-squared tests to examine the relationship of each domain of KAP (low, moderate, high) with dengue seropositivity. Kendall rank correlation tests were used to assess the associations between each pair of KAP domains (i.e., K and A, K and P, A and P). The associations between sociodemographic covariates and seropositivity were assessed through univariate regression. A multivariable regression model was then developed to assess associations between risk of seropositivity and potential predictors of risk including KAP, rainy season, age, race, average monthly household income, household water storage method, household's recent dengue episode and dengue death history, and the education level and occupation status of the household's principal adult respondent. The selection of variables was guided by hypothesized causal associations and confounders through literature review and direct acyclic graphs. To address any overdispersion by clustering we considered a random effects model and a quasibinomial model but the appropriateness of the random effects model was limited by the small cluster sizes and small number of clusters and the quasibinomial model dispersion parameter was 1.0 [25], indicating little evidence of overdispersion. For both univariable and multivariable regression models, relative risks (RR) and 95% confidence intervals were estimated based on Zou's modified Poisson regression approach [26]; an alternative to logistic regression that directly estimates relative risk, rather than odds ratios, and is appropriate when the outcome is common [27].

Children's ages were divided into two groups: 2–5 years old and 6 to 12 years old. Children's race was classified into five categories, based upon the classification used by the *Instituto Brasileiro de Geografia e Estatística* which is used for the Brazilian census: *Branco* (White), *Pardo ou moreno* (Multiracial), *Preto* (Black), *Asiático* (Asian), and *Indígena* (Indigenous) [28]. Race was reduced to two categories, multiracial and white vs. other, for the multivariable regression to ensure sufficient representation in the race categories.

To consider the climate factor, we separated the children according to the date their household was surveyed and the date the blood sample was collected (both occurring on the same day), using cumulative monthly rainfall [29]. We distinguished between households with no potable water storage and sealed individual water tanks from households with open water storage methods (e.g., buckets, pots) [30]. Households were considered to have a dengue history if they experienced a recent dengue episode or the death of a family member due to dengue. Household education level of the adult respondent was categorized based upon completion of elementary school as elementary school is essential for an individual's literacy. There were two categories of occupation status, respondents who had paid employment at the time of the

questionnaire vs being unemployed, retired, or other (e.g., refused to answer). Households' monthly income were categorized as low (<500 Brazilian real), moderate (>500 or <2500 Brazilian real), or high (>2500 Brazilian real) based on Fortaleza's cost of living and the average household income of the resident population from the Ceara state [28,31]. MICE imputation was carried out for the multivariable regression analysis to impute missing household income and children's race.

All statistical analyses were conducted using R version 4.2.0.

## Results

### Characteristics of participating households and children

There were 392 households and 483 children who provided a blood sample of sufficient quality for this analysis. The mean age of participating children was 6.5 years with an almost equal proportion of boys and girls participating in the study (Table 1). The mean age (SD) of household principal adult respondents was 29 years with the majority being female (n = 345, 88%). Children were most often reported as being multiracial/white (n = 434, 90%). Parental respondents had a high completion rate of elementary school (n = 276, 70%) and most households had a moderate household income (n = 283, 72%). Household and children's sample size per borough,and characteristics of the study population after imputation can be found in S2 and S3 Appendices respectively.

Table 2 shows crude seroprevalence, seroprevalence differences and ratios for children by individual characteristics. There were 121 seropositive samples and 362 seronegative samples, resulting in an estimated crude seroprevalence of 25.1% (95% CI: 21.3% - 29.2%).

### Knowledge, attitudes, practices and overall KAP

Only five percent (n = 20) of households achieved over 75% for overall KAP, while most households achieved scores between 50% and 75% (n = 271, 69%), and 101 households scored lower than 50% of the total score (26%). Table 3 shows the result of the Kendall rank correlation tests between each KAP domain and the result of Chi-squared tests with children's seropositivity. The results of the Kendall rank correlation tests and Chi-squared tests demonstrate that the three domains have a statistically significant positive association with each other. However, in univariate tests, there was no evidence of an association between children's seropositivity and the household's level of each individual domain (knowledge, practice, or attitude) or with the overall KAP measure.

For each KAP question, the percentage score was calculated by dividing the score of each household achieved by the maximum points of each question (S4 Appendix). Out of the three domains, the knowledge domain had the highest mean percentage score of 74%, followed by the attitude domain and then the practice domain at 63% and 39%, respectively. While all questions in the knowledge domain have a mean percentage score over 65%, the mean percentage score varies largely among questions in the attitude and practice domains.

### Demographic and socioeconomic determinants of dengue seropositivity in children

In the multivariable regression model, the parental education level, households' KAP level, age of the child, and household water storage method were shown to be significantly associated with the risk of dengue seropositivity in children (Table 4).

Among all included predictors, children's age was most strongly associated with the risk of dengue seropositivity, which was significantly higher among children aged 6–12 years

**Table 1. Sociodemographic Characteristics of Children and Households, total children participants N = 483.**

| Children | Total N (%) |
|---|---|
| **Child Age, year** | |
| Mean (SD) | 6.5 (2.7) |
| Median [Min, Max] | 6.0 [2.0, 12.0] |
| 2-5 | 181 (50%) |
| 6-12 | 181 (50%) |
| **Child Race** | |
| Multiracial/White | 434 (89.9%) |
| Other | 49 (10.1%) |
| **Child Gender** | |
| Female | 222 (46.0%) |
| Male | 261 (54.0%) |
| **School Attendance** | |
| Yes | 431 (89.2%) |
| No | 50 (10.4%) |
| Refused | 2 (0.4%) |
| **Households** | **Total (N=392)** |
| **Household Monthly Income (BRL)\*** | |
| Low | 21 (5.4%) |
| Moderate | 254 (64.8%) |
| High | 79 (20.2%) |
| Nonresponse | 38 (9.7%) |
| **Respondent Education** | |
| Lower than Elementary School | 116 (29.6%) |
| Elementary School or Higher | 276 (70.4%) |
| **Respondent Gender** | |
| Female | 348 (88.8%) |
| Male | 44 (11.2%) |
| Does not know | 3 (0.8%) |
| **Respondent Race** | |
| Black | 28 (9.7%) |
| Multiracial | 285 (72.7%) |
| White | 55 (14.0%) |
| Asian | 10 (2.6%) |
| Other | 4 (1.0%) |
| **Occupation** | |
| Retired/Unemployed/Other | 202 (51.5%) |
| Employed | 190 (48.5%) |
| **Household KAP** | |
| Low | 101 (25.8%) |
| Moderate | 271 (69.1%) |
| High | 20 (5.1%) |

\*BRL = Brazilian real

compared to children aged 2–5 years old (RR: 6.1, 95% CI: 3.5–10.7, p<0.001). Children of parents with higher education levels, who at least completed elementary school, were less likely to be seropositive (RR: 0.7, 95% CI: 0.5–0.9, p = 0.005). There were no statistically significant

**Table 2. Crude Seroprevalence and Seroprevalence Differences and Ratios for Children by Individual Characteristics.**

|  | Total No. | Seroprevalence, % | 95% CI | Crude prevalence ratio (95% CI) |
|---|---|---|---|---|
| **Total** | 483 | 25.1 | 21.3–29.2 | NA |
| **Child Age, year** |  |  |  |  |
| 2–5 | 193 | 6.2 | 3.4–10.9 | 1 [Reference] |
| 6–12 | 290 | 37.6 | 32.0–43.5 | 6.1 (3.4–10.7) |
| **Child Race*** |  |  |  |  |
| Multiracial/White | 440 | 25.0 | 21.0–29.3 | 1 [Reference] |
| Other | 43 | 25.6 | 13.5–41.2 | 1.0 (0.5–2.1) |
| **Child Gender** |  |  |  |  |
| Female | 222 | 27.0 | 21.4–33.5 | 1 [Reference] |
| Male | 261 | 23.3 | 18.5–29.1 | 0.8 (0.5–1.2) |
| **School Attendance** |  |  |  |  |
| Yes | 431 | 27.4 | 23.3–31.9 | 1 [Reference] |
| No | 50 | 4.0 | 0.7–14.9 | 0.1 (0.0–0.5) |
| Refused | 2 | 50.0 | 9.5–90.5 | 2.7 (0.2–42.8) |

* "Does not know" responses for Child Race were imputed

associations detected between dengue seropositivity and monthly household income, principal adult's occupation, children's race, household's dengue history or rainy season. Children living in households without potable water storage or using sealed water tanks were found to be significantly associated with a lower risk of dengue seropositivity (RR: 0.7, 95% CI: 0.5–1.0, p = 0.038). Notably, our result showed that higher values of KAP were associated with an increased risk of dengue seropositivity (RR: 2.1, 95% CI: 1.1–4.0, p = 0.024).

## Discussion

The results of our study demonstrated that one quarter of the participating children were seropositive for dengue and the results identified certain socioeconomic and demographic characteristics that were associated with the risk of dengue seropositivity. The results also suggest that KAP was not a protective predictor of dengue seropositivity risk in our cross-sectional study, as it was positively associated with seropositivity.

**Table 3. Kendall Rank Correlation and Chi-Square Test Summary.**

|  | tau[1] | p-value | |
|---|---|---|---|
| Attitude vs. Practice | 0.12 | 0.008 | |
| Practice vs. Knowledge | 0.13 | 0.003 | |
| Knowledge vs. Attitude | 0.13 | 0.002 | |
|  | **Chi-square** | **df[2]** | **p-value** |
| Attitude vs. Serology Result | 2.32 | 2 | 0.313 |
| Practice vs. Serology Result | 5.43 | 2 | 0.066 |
| Knowledge vs. Serology Result | 1.63 | 2 | 0.442 |
| Overall KAP vs. Serology Result | 3.75 | 2 | 0.153 |

[1]tau = Kendall's tau statistic

[1]tau = Kendall's tau statistic

[1]tau = Kendall's tau statistic

[2]df = Degree of freedom

**Table 4. Multivariable Regression Model of Sociodemographic Predictors of Children's Serology Results.**

| Characteristic | Levels | RR[1] | 95%CI[1] | p-value |
|---|---|---|---|---|
| Household KAP | Low | REF | REF | REF |
| | Moderate | 1.23 | [0.86, 1.75] | 0.267 |
| | High | 2.11 | [1.11, 4.02] | 0.024* |
| Respondent's Education | Less than Elementary School | REF | REF | REF |
| | Elementary School or Higher | 0.65 | [0.48, 0.87] | 0.004** |
| Child Age | Younger Group (2–5 years old) | REF | REF | REF |
| | Older Group (6–12 years old) | 6.09 | [3.47, 10.69] | < 0.001*** |
| Child Race | Other | REF | REF | REF |
| | Multiracial/White | 1.03 | [0.63, 1.69] | 0.915 |
| Water Storage Method | Other Water Storage Method | REF | REF | REF |
| | Sealed Water Tank/No Water Storage | 0.73 | [0.55, 0.98] | 0.038* |
| Respondent's Occupation | Retired/Unemployed/Other | REF | REF | REF |
| | Employed | 0.75 | [0.56, 1.02] | 0.065 |
| Household Dengue Experience | Yes | REF | REF | REF |
| | No | 0.80 | [0.58, 1.10] | 0.18 |
| Household's Monthly Income | Low | REF | REF | REF |
| | Moderate | 0.96 | [0.56, 1.65] | 0.89 |
| | High | 1.26 | [0.68, 2.30] | 0.460 |
| Rainy Season | No | REF | REF | REF |
| | Yes | 1.11 | [0.84, 1.48] | 0.462 |

[1] RR = Relative Risk, CI = Confidence Interval

The overall dengue seroprevalence was comparable to other dengue seroprevalence studies of children in dengue-endemic settings [32,33]. We also found that the seroprevalence varied widely between different sociodemographic factors such as race, sex, and age. Black and white children had the highest seroprevalence compared to multiracial and other groups although given the low number of seropositive children in certain racial groups, these differences should be interpreted with caution. Associations between racial and ethnic minority status and dengue seropositivity have been found by other studies [16,34,35], and race is often related to increased vulnerability of disease due to social conditions including housing, municipal services, and education [36].

Girls were estimated to have a higher dengue seroprevalence compared to boys, although the difference was not statistically significant. There are a mixture of findings in terms of gender differences in dengue incidence, some have found greater male dengue incidence among both children and adults which is likely due to gender-related differences in exposure such as time away from home and participation in outdoor games [36–39], while others have found no differences [40,41]. In our study, we also found that children's age was positively associated with seropositivity, which is comparable to many similar studies [32,42,43]. Considering IgG remains detectable in blood for many years [44], children in the older group (6–12 years old) had a longer time at risk of infection compared to younger children. Previous studies have determined that the majority of dengue virus transmission occurs within communities or schools, school attendance could play an important role in the progression of dengue epidemics given the increases in the dengue infection rate following the beginning of the school year [38,45,46].

Children were 35% less likely to be seropositive if the parent/guardian respondent had at least completed elementary school, compared to those children whose parent/guardian had

lower educational attainment. Multiple studies have demonstrated that education was positively associated with the use and promotion of dengue prevention, such as ability to take actions of control, acceptability of prevention measures and educating family members and neighbors [47,48]. Our study also suggests that water storage methods are important in terms of risk of dengue seropositivity in children, which has been found by several other studies [49–51]. Open containers and unsealed reservoirs can act as important breeding sites for *Aedes* mosquitos [12,49,51], while piped water and sealed reservoirs can reduce the presence of *Aedes* mosquitos significantly [12,49].

An interesting finding from our study is that children living in households with high KAP levels had a higher relative risk of dengue seropositivity. One hypothesis for this finding is related to our outcome measure and an important limitation of cross-sectional data—a lack of temporality. We did not measure active dengue infections but measured dengue seropositivity using IgG, which can persist in the blood for several years [44]. It is plausible that this inverse association is due to a mismatch of temporality between our outcome and exposure of interest.

Cohort and case-control studies have found participants who had experience with dengue had a significantly higher knowledge score, as they are likely to seek for information when encountering the disease or have more chances to learn from healthcare staff when receiving treatments [52–54]. Among children with previous dengue, parents/guardians have been more likely to perceive dengue as a serious concern [46]. Studies conducted in Indonesia and Colombia have shown that adults who have experienced dengue are also more likely to have a positive attitude toward preventing it [55,56]. In terms of the impact of dengue experience on prevention practices, a study from Malaysia found age and dengue history are the main determinants that influence a high practice level [57], which is consistent with another study's result where previous dengue infections were positively associated with dengue prevention practices [58]. Therefore, if a child had a previous dengue infection, it is likely that this experience improved the overall KAP in the household [52,58].

Participating households demonstrated a good understanding of vector control and dengue transmission, prevention, and symptoms, as evidenced by the relatively high overall mean percentage score in the knowledge domain compared to the knowledge level observed in previous studies conducted in dengue pandemic areas [20,59,60]. This phenomenon can likely be attributed to the high prevalence of dengue fever in Brazil and exposure to dengue prevention education campaigns and related information.

In terms of attitudes, a fair overall mean score was observed. Almost all participating respondents believe they were at-risk of dengue and that dengue can be prevented although the majority of participants lacked confidence in the community's dengue prevention capabilities. This attitude could be due to several factors, which include a lack of support from government and community services [61–63], and disillusionment about household and community agency [64,65]. There was poor level of household-level vector control practices and in certain studies, it was demonstrated that knowledge was not associated with any significant behavior change [66,67]. Given the low confidence in community prevention ability among the study population, community engagement activities may be necessary to deliver key messages on dengue and to work with the community to identify acceptable vector control interventions [66,68–70]. This would ensure the sustainability of long-term community dengue prevention interventions.

## Limitations

The study has several limitations. The main limitation is the cross-sectional design with exposure and outcome measured at the same time, making it impossible to establish temporality in

the associations. KAP can change substantially over time and in response to certain events, such as previous dengue infection. There is likely selection bias, as the response rate varied by clusters and decreased throughout the study due to the political context. Information bias could have occurred with the questionnaire data collection given that it was based on self-reported information, particularly for the KAP questions, although we assume that the impact would be non-differential. While the diagnostic test Panbio Dengue IgG Indirect ELISA is proven to be highly reliable and showed high agreement with HAI [23,24], there is a possibility that some participants were misclassified in terms of seropositivity (false negative) with 95.2%-97.9% sensitivity for secondary infections (77.1% sensitivity for primary) and 93.4%-100% specificity [22,23,71]. This may have led to misclassification of serostatus. Confounding bias could have occurred via residual confounding as we reduced the number of categories within certain variables to improve the precision of our results. In terms of external validity or generalizability, the study population was representative of Fortaleza in terms of race and reported household income, although may not have been representative for study characteristics [28,31].

## Conclusion

We found certain demographic and socio-economic characteristics, particularly children's age, parent's education, and water storage method were associated with children's risk of seropositivity. Further research is needed to identify barriers that influence attitudes and practice among the study population and to understand if and how community mobilization is an effective approach for dengue control. We also identified an important limitation of using KAP measures for cross-sectional studies for dengue infection: KAP could change due to previous infection experience, which is likely relevant for other infectious diseases. Despite this limitation of using KAP as a potential predictor of dengue seropositivity in cross-sectional studies, it remains useful for identifying gaps in knowledge, attitude, and practice that can be used to inform public health measures, such as education campaigns. Measures of social acceptability of interventions should also be considered for inclusion in similar studies [72], as it would provide an indication of likelihood of adoption of interventions, which provides additional insight for different dengue interventions.

## Supporting information

**S1 Appendix. KAP Survey Questionnaire and Scoring.**
(DOCX)

**S2 Appendix. Household and Children's Sample Size per Borough.**
(DOCX)

**S3 Appendix. Sociodemographic Characteristics of Children and Households After Imputation.**
(DOCX)

**S4 Appendix. Mean Percentage Score of KAP Questions with 95% Confidence Interval.**
(DOCX)

## Author Contributions

**Conceptualization:** Laura Pierce, Valéry Ridde, Kate Zinszer.

**Data curation:** Yang Zhang, Monica Zahreddine, Kellyanne Abreu, Mayana Azevedo Dantas.

**Formal analysis:** Yang Zhang, Katia Charland, Laura Pierce.

**Funding acquisition:** Valéry Ridde, Kate Zinszer.

**Investigation:** Monica Zahreddine, Kellyanne Abreu, Mayana Azevedo Dantas.

**Methodology:** Yang Zhang, Katia Charland, Kate Zinszer.

**Project administration:** Monica Zahreddine, Kellyanne Abreu, Mayana Azevedo Dantas, Kate Zinszer.

**Supervision:** Monica Zahreddine, Kellyanne Abreu, Mayana Azevedo Dantas, Kate Zinszer.

**Validation:** Yang Zhang, Monica Zahreddine.

**Visualization:** Yang Zhang.

**Writing – original draft:** Yang Zhang, Kate Zinszer.

**Writing – review & editing:** Yang Zhang, Monica Zahreddine, Katia Charland, Laura Pierce, Valéry Ridde, Kate Zinszer.

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
