## [Decision Letter · Decision Letter 0]

21 Mar 2023

Dear Dr. Zinszer,

Thank you very much for submitting your manuscript "A cross-sectional study on knowledge, attitude and practice (KAP) and risk factors for dengue seropositivity among children in Fortaleza, Brazil." for consideration at PLOS Neglected Tropical Diseases. As with all papers reviewed by the journal, your manuscript was reviewed by members of the editorial board and by several independent reviewers. In light of the reviews (below this email), we would like to invite the resubmission of a significantly-revised version that takes into account the reviewers' comments. 

We cannot make any decision about publication until we have seen the revised manuscript and your response to the reviewers' comments. Your revised manuscript is also likely to be sent to reviewers for further evaluation.

Sincerely,

Ran Wang, M.D.

Academic Editor

Alvaro Acosta-Serrano

Section Editor

Reviewer's Responses to Questions

**Key Review Criteria Required for Acceptance?**

**Methods**

-Are the objectives of the study clearly articulated with a clear testable hypothesis stated?

-Is the study design appropriate to address the stated objectives?

-Is the population clearly described and appropriate for the hypothesis being tested?

-Is the sample size sufficient to ensure adequate power to address the hypothesis being tested?

-Were correct statistical analysis used to support conclusions?

-Are there concerns about ethical or regulatory requirements being met?

Reviewer #1: 1. The sample size for serology is rather small (<500 children) and I note that in the methods further down the intention was to do ten times more clusters and people. Whilst we all appreciate the pandemic problems, it unfortunately highlights the problem of the sample size. One would presume that some power calculations had been done for the larger wished for sample size, but the study is now clearly under-powered. It is also unclear whether the study that aimed for a 10 times greater sample size is being pursued or that simply these were the only samples that could be obtained during the project time frame. Could the authors perhaps do a retrospective power calculation based on their initial power calculation to reveal what power the small achieved number of samples could be expected to achieve?

2. It would be useful to indicate the age of the children in parentheses (it would seem from further down that children assessed for serology are 2-12 years old). Why this age group and not for example up to 18 years of age. Are these groups based on previous seroprevalence data? And which thus provide the basis for the sample size calculations? 

3. As mentioned above, this was meant to be a larger study and part of a cluster randomized control trial published by the authors in Trials. Can the authors explain how a cluster was defined, insofar as here they achieved sampling from 4 clusters.

4. I fail to see an ethics statement about written and oral (?) informed consent for parents/guardians and their children, although I see further down the ethical approval by the pertinent ethics committees.

5. The justification behind the methodology and scoring of the KAP study is not clearly defined and the reference given from a Saudi Covid study does not seem to be adequate as a reference methodology. There are many KAP methodology papers that can be referred to. A recent paper with very many pertinent references can be found here: https://doi.org/10.1016/j.envres.2020.110509

6. As mentioned above, there lacks a sample size calculation.

7. Elution of the immunoglobulins from the bloodspot for the PanBio ELISA. Can the authors write how they eluted the IgGs from the dried blood spot?

8. More information on the KAP questions: there are very few and so perhaps these can be spelled out and justified.

9. Statistical analysis: as one might expect KAP variables to be somewhat correlated, were there any attempts to look at co-linearity and perhaps use of Principle components analyses to generate novel variables that are comprised of like KAP variables?

10. As there were more children than households, there must have been multiple children from a single household. How was this taken into account in the analyses?

Reviewer #2: (No Response)

Reviewer #3: The aim and objectives are clear. The methods are well written and largely reproducible. The authors acknowledge that the sample size was not met due to the pandemic (would probably specify Covid-19) and instability in the region. Suggest to specify that full details on the methods / study population can be found in Zinszer et al., (2020)

Also to specify how a cluster is defined and selected. Further, it would be useful to know whether the clusters included were randomly distributed throughout the city (or concentrated in a specific area etc.). 

It would also be useful for other researchers to include the KAP questionnaire in the supplementary information. The sample size calculation (given the reduction in sample size is not presented).

**Results**

-Does the analysis presented match the analysis plan?

-Are the results clearly and completely presented?

-Are the figures (Tables, Images) of sufficient quality for clarity?

Reviewer #1: 11. The results are very interesting, especially given the small sample size.

12. It would be very useful to have more information about the KAP results at a single variable level. Just presenting a KAP score is too brief.

Reviewer #2: (No Response)

Reviewer #3: The results are presented well. Multivariable regression models are more commonly constructed as logistic regression and odds ratios presented. Does the multivariate model account for within-cluster correlation? It might be better to perform logistic regression with random effects for the multivariate risk factor analysis. 

It would probably be useful for the reader to have the KAP results added to supplementary information so that they can see which parts scored high / low.

**Conclusions**

-Are the conclusions supported by the data presented?

-Are the limitations of analysis clearly described?

-Do the authors discuss how these data can be helpful to advance our understanding of the topic under study?

-Is public health relevance addressed?

Reviewer #1: 13. Likewise the discussion on the KAP findings is almost non-existent. As stated perhaps the respondents have higher KAP scores because they are more confronted by dengue. So dengue occurrence drives KAP. However, whilst history of dengue might improve knowledge, did it improve attitude and practices?

Reviewer #2: (No Response)

Reviewer #3: The discussion is well written and acknowledges study limitations. It would be good to discuss the impact of the pandemic / reduction in target numbers again in the study limitations. The discussion focuses on seropositivity and associations with KAP, it would also be useful to include some general results/recommendations regarding KAPs and the study population. So were the KAP scores in the study population high/low – was there particularly areas which had low awareness which could be targeted for improvement.

**Editorial and Data Presentation Modifications?**

Reviewer #1: (No Response)

Reviewer #2: (No Response)

Reviewer #3: (No Response)

**Summary and General Comments**

Reviewer #1: An interesting pilot study on a subject of interest, namely relating KAP to dengue seropositivity. However, through clearly collateral covid-associated damage, the samle sizes achieved are too small to be very meaningful. As a pilot study this is OK. 

1. This is a study relating KAP, socioeconomics and dengue seropositivity. A recent article in PNTD in India looked exactly at this question and including the precise aspects on access to drinking water stated in the present study. I would suggest the authors read this article and the references cited within to put their study into what has been recently published on this: doi: 10.1371/journal.pntd.0009024.

Abstract

2. Abstract line 2 : It reads « Children are vulnerable to infection with the dengue virus and the risk of severe dengue disease is the highest among infants and children, particularly those who have been previously infected with a different dengue serotype.”

This isn’t actually true and it depends very much on the force of infection. Even in many endemic countries the average age of first infection can be as late as 10 years old and the increased disease severity following a secondary infection thus occurs later into young adulthood. The key point is that age-severity is very much context dependent.

3. The section introducing KAP is a little limited and lacks references with respect to dengue. The authors write that there have been several studies on KAP and dengue, not one is cited.

Reviewer #2: In this manuscript, Zhang and colleagues performed a cross-sectional study of dengue seroprevalence in Fortaleza, Brazil between November 2019 and February 2020. They examined the association between characteristics of certain study populations and risk of dengue seropositivity. The authors found that children’s age was most strongly associated with the risk of dengue seropositivity. I have a few suggestions:

(1) It’s not very clear why the authors collected these data for only a single dengue season in the study location. The seasonality of dengue outbreaks can be driven by many factors, including climate conditions and pre-existing population immunity. The authors may wish to examine the influence of these factors on the observed seroprevalence. 

(2) The authors may also wish to examine the difference in seroprevalence between the study period and the same period in previous years.

Reviewer #3: (No Response)
---

## [Decision Letter · Decision Letter 1]

25 Jul 2023

Dear Dr. Zinszer,

Thank you very much for submitting your manuscript "Knowledge, attitude and practice (KAP) and risk factors on dengue fever among children in Brazil, Fortaleza: A cross-sectional study" for consideration at PLOS Neglected Tropical Diseases. As with all papers reviewed by the journal, your manuscript was reviewed by members of the editorial board and by several independent reviewers. In light of the reviews (below this email), we would like to invite the resubmission of a significantly-revised version that takes into account the reviewers' comments. 

We cannot make any decision about publication until we have seen the revised manuscript and your response to the reviewers' comments. Your revised manuscript is also likely to be sent to reviewers for further evaluation.

Sincerely,

Ran Wang, M.D.

Academic Editor

Álvaro Acosta-Serrano

Section Editor

Reviewer's Responses to Questions

**Key Review Criteria Required for Acceptance?**

**Methods**

-Are the objectives of the study clearly articulated with a clear testable hypothesis stated?

-Is the study design appropriate to address the stated objectives?

-Is the population clearly described and appropriate for the hypothesis being tested?

-Is the sample size sufficient to ensure adequate power to address the hypothesis being tested?

-Were correct statistical analysis used to support conclusions?

-Are there concerns about ethical or regulatory requirements being met?

Reviewer #2: (No Response)

Reviewer #3: A map of the distribution of the clusters (especially if dengue is influenced by environmental conditions would be useful).

Reviewer #4: - The study is a baseline survey of cRCT, which supposed to be conducted in 68 clusters that were selected from 3,020 clusters in Fortaleza, Brazil. However, authors only managed to collect data from four clusters and partial data from 8 clusters. Are the 4 clusters representative for 3020 clusters? 

- Authors mentioned that they collected partial data from eight clusters, what do they mean by partial data?

- A total of 392 households and 483 participant children were enrolled in the study. The sample size may be reasonable if authors used simple random sampling, but the sampling method was cluster sampling, which requires higher sample size.

- It is not clear how did authors selected households in the selected clusters.

- Are all children in the households were enrolled?

**Results**

-Does the analysis presented match the analysis plan?

-Are the results clearly and completely presented?

-Are the figures (Tables, Images) of sufficient quality for clarity?

Reviewer #2: (No Response)

Reviewer #3: If the within-cluster correlation is not controlled for – I would add the justification for this somewhere in the manuscript.

Reviewer #4: The presentation of results is poor:

- Table 1 should be shortened

- Table 2 is not well presented: I suggest one column for sample size N, one column for positive n(%), one column for 95%CI and one column for prevalence ratio.

- Table 3 is not well presented, adding the descriptive data may make the table more informative

- Do tables 4 – 6 add much?

- Table 7: what about adding knowledge attitudes and practices presented in table 3 to the risk factors and use both Bivariate and multivariable analysis. In addition, the descriptive data makes the table more informative.

**Conclusions**

-Are the conclusions supported by the data presented?

-Are the limitations of analysis clearly described?

-Do the authors discuss how these data can be helpful to advance our understanding of the topic under study?

-Is public health relevance addressed?

Reviewer #2: (No Response)

Reviewer #3: Further discussion of the ELISA test performance would be useful – has the test been sufficiently evaluated?

Reviewer #4: (No Response)

**Editorial and Data Presentation Modifications?**

Reviewer #2: (No Response)

Reviewer #3: (No Response)

Reviewer #4: (No Response)

**Summary and General Comments**

Reviewer #2: (No Response)

Reviewer #3: The study is much improved, just a few minor suggestions.

Reviewer #4: (No Response)
---

## [Decision Letter · Decision Letter 2]

12 Sep 2023

Dear Dr. Zinszer,

We are pleased to inform you that your manuscript 'Knowledge, attitude and practice (KAP) and risk factors on dengue fever among children in Brazil, Fortaleza: A cross-sectional study' has been provisionally accepted for publication in PLOS Neglected Tropical Diseases.

Best regards,

Ran Wang, M.D.

Academic Editor

Álvaro Acosta-Serrano

Section Editor

Reviewer's Responses to Questions

**Key Review Criteria Required for Acceptance?**

**Methods**

-Are the objectives of the study clearly articulated with a clear testable hypothesis stated?

-Is the study design appropriate to address the stated objectives?

-Is the population clearly described and appropriate for the hypothesis being tested?

-Is the sample size sufficient to ensure adequate power to address the hypothesis being tested?

-Were correct statistical analysis used to support conclusions?

-Are there concerns about ethical or regulatory requirements being met?

Reviewer #4: (No Response)

**Results**

-Does the analysis presented match the analysis plan?

-Are the results clearly and completely presented?

-Are the figures (Tables, Images) of sufficient quality for clarity?

Reviewer #4: (No Response)

**Conclusions**

-Are the conclusions supported by the data presented?

-Are the limitations of analysis clearly described?

-Do the authors discuss how these data can be helpful to advance our understanding of the topic under study?

-Is public health relevance addressed?

Reviewer #4: (No Response)

**Editorial and Data Presentation Modifications?**

Reviewer #4: (No Response)

**Summary and General Comments**

Reviewer #4: (No Response)

PLOS authors have the option to publish the peer review history of their article (what does this mean?). If published, this will include your full peer review and any attached files.

Reviewer #4: No

---

## [Editor Report · Acceptance letter]

20 Sep 2023

Dear Dr. Zinszer,

We are delighted to inform you that your manuscript, "Knowledge, attitude and practice (KAP) and risk factors on dengue fever among children in Brazil, Fortaleza: A cross-sectional study," has been formally accepted for publication in PLOS Neglected Tropical Diseases.

Best regards,

Shaden Kamhawi

co-Editor-in-Chief

Paul Brindley

co-Editor-in-Chief
